# Evaluation of Antiviral Activity of Cyclic Ketones against Mayaro Virus

**DOI:** 10.3390/v13112123

**Published:** 2021-10-21

**Authors:** Luciana S. Fernandes, Milene L. da Silva, Roberto S. Dias, Marcel S. da S. Lucindo, Ítalo E. P. da Silva, Cynthia C. Silva, Róbson R. Teixeira, Sérgio O. de Paula

**Affiliations:** 1Molecular Immunovirology Laboratory, General Biology Department, Universidade Federal de Viçosa, Viçosa 36570-900, MG, Brazil; luciana.fernandes@ufv.br (L.S.F.); rosousa318@gmail.com (R.S.D.); marcelseverinos@gmail.com (M.S.d.S.L.); italoesposti@gmail.com (Í.E.P.d.S.); 2Microbiology Department, Universidade Federal de Viçosa, Viçosa 36570-900, MG, Brazil; milenelopesdasilva@yahoo.com.br (M.L.d.S.); ccanedosilva@gmail.com (C.C.S.); 3Chemistry Department, Universidade Federal de Viçosa, Viçosa 36570-900, MG, Brazil

**Keywords:** mayaro virus, cyclic ketones, xanthenodiones, antivirals, arbovirus

## Abstract

Mayaro virus (MAYV) is a neglected arthropod-borne virus found in the Americas. MAYV infection results in Mayaro fever, a non-lethal debilitating disease characterized by a strong inflammatory response affecting the joints and muscles. MAYV was once considered endemic to forested areas in Brazil but has managed to adapt and spread to urban regions using new vectors, such as *Aedes aegypti*, and has the potential to cause serious epidemics in the future. Currently, there are no vaccines or specific treatments against MAYV. In this study, the antiviral activity of a series of synthetic cyclic ketones were evaluated for the first time against MAYV. Twenty-four compounds were screened in a cell viability assay, and eight were selected for further evaluation. Effective concentration (EC_50_) and selectivity index (SI) were calculated and compound 9-(5-(4-chlorophenyl]furan-2-yl)-3,6-dimethyl-3,4,5,6,7,9-hexahydro-1H-xanthene-1,8(2))-dione (9) (EC_50_ = 21.5 µmol·L^−1^, SI = 15.8) was selected for mechanism of action assays. The substance was able to reduce viral activity by approximately 70% in both pre-treatment and post-treatment assays.

## 1. Introduction

Mayaro virus (MAYV, *Togaviridae* family, *Alphavirus* genus) is an arthropod-borne virus in Central and South America [1]. The first report of MAYV occurred in 1954 when the virus was isolated from blood samples of five rural workers in Mayaro County, a southeastern region of Trinidad [2]. In the following year, the first cases of infections by MAYV were reported in Brazil, where the virus was isolated from six patients (also rural workers) [3]. Since then, MAYV has been responsible for sporadic outbreaks in rural and forested areas of the country, mainly in the North and Central regions [4]. MAYV is part of the Semliki complex, a serological group within the Alphavirus genus that also includes Chikungunya virus (CHIKV), Bebaru virus (BEBV), O’nyong o’nyong virus (ONNV), Semliki Forest virus (SFV), Getah virus (GETV), Ross River virus (RRV), and Una virus (UNAV). Members of this group share antigenic sites that can result in cross-reactivity and lead to misdiagnosis in endemic areas where they co-circulate [4,5].

MAYV is maintained in a sylvatic cycle using *Haemagogus janthinomys* mosquitoes as primary vectors and a variety of mammalian species as reservoirs such as rodents, birds, and non-human primates [6]. Although considered endemic to rural areas, MAYV has been detected in urban areas in Brazil (frequently in the Mato Grosso State, central region), potentially being transmitted by urbanized vectors such as *Aedes* spp and *Culex* spp [7,8,9,10,11]. These arthropod species are distributed nationwide, and the fact that most of the population has never been in contact with MAYV considerably increases the potential of viral emergence and the threat of large epidemics. Moreover, it is known that *Aedes albopictus* is resistant to lower temperatures, which can facilitate its spread to temperate regions where the virus is non-endemic [12]. As confirmed by an epidemiological study, Brazil has the highest number of MAYV reported cases (1304). Most reports concentrate in the Northern region, in the Legal Amazon [12]. Continuous deforestation and unplanned urbanization of forested areas also contribute to the virus expansion and emergence [13]. This represents a severe concern for the public health system, especially in Brazil, where there is a co-circulation of other arthropod-borne viruses, such as Dengue virus, Chikungunya virus, and Zika virus, causing an underestimation of infection rates [14,15]. 

MAYV is the etiologic agent of Mayaro fever, a febrile illness characterized by a strong inflammatory immune response with clinical manifestations such as fever, headaches, myalgia, arthralgia/arthritis, rash, vomiting, and diarrhea. The most striking feature of Mayaro fever is the development of debilitating long-term arthritis that can last for several months. Although non-lethal, Mayaro fever can severely impact an individual’s life quality. Thus, the similarity of symptoms to other arboviral diseases, such as chikungunya and dengue fever, often impairs correct diagnosis [6,16,17].

There are no available vaccines for preventing MAYV infection, although a few studies have been conducted on this matter [18,19,20]. Specific treatment against MAYV is also still lacking, and the search for compounds with antiviral activity is necessary considering the possibility of an epidemic scenario. Nonetheless, studies have been published investigating various natural and synthetic compounds for anti-MAYV activity [21,22,23,24,25,26]. Mayaro fever is usually treated aiming at symptom relief with analgesics and non-steroidal anti-inflammatory drugs [4].

Cyclic ketones, either natural or synthetic, are compounds endowed with valuable biological activities. For example, thujone, carvone, and thymoquinone (Figure 1) are natural monoterpene ketones and present, respectively, antiviral [27,28], antioxidant [29], and antifungal [30] activities. The triketones sulcotrione and mesotrione are modern herbicides used to control several weed species in different crops [31]. Indan-1,3-diones bearing arylidene groups display leishmanicidal and cytotoxic activities [32]. Other important cyclic ketones are the xanthenodiones [or 1,8-dioxo-octahydro-xanthenes] characterized by the presence of a pyran-nucleus fused to a cyclohexe-2-one ring on both sides. These compounds present a variety of biological functions, including anti bactericidal, leishmanicidal, antifungal, antitumoral, and tripanocidal activities [33,34]. Xanthenodiones resemble xanthones, a class of natural compounds known for their numerous pharmacological applications including antiviral, antimicrobial, hepatoprotective, anticancer, antioxidant, and anti-inflammatory applications [35]. These examples show the diverse biological profile presented by cyclic ketones that can be explored in the pharmaceutical and agrochemical fields. 

Our research group has previously demonstrated that diketones bearing arylidene functionalities present a significant inhibitory effect on NS2B-NS3 West Nile virus protease [36]. Additionally, we have proved the antiviral activity of cyclic ketones against the Zika virus [37]. In continuation of our investigations toward discovering potential antiviral compounds, in this study, we evaluated twenty-four cyclic ketones (twenty-one xanthenodiones, one diketone with arylidene moiety, and two tetraketones) for their antiviral activity against MAYV.

## 2. Materials and Methods

### 2.1. General Procedure for Preparation of Cyclic Ketones 1–24

A round-bottomed flask (25 mL) was charged with a 1,3-diketone (2.00 mmol), aldehyde (1.00 mmol), and ZrOCl_2_·8H_2_O (12.0 mg, 2 mol%). The mixture was stirred at 85 °C and the progress of the reaction was monitored by TLC analysis. After completion of the reaction, the mixture was cooled down to room temperature. Thereafter, it was added 50 mL of dichloromethane and the mixture was kept under stirring for about 30 min. Then, the catalyst, which is insoluble in dichloromethane, was separated by filtration. After that, 50 mL of ethanol was added to the filtrated and the system was kept undisturbed for crystallization. Detailed description of cyclic ketones synthesis was described previously [27,37].

### 2.2. Cell Culture and Viral Stock

Vero cells (African green monkey kidney, ATCC CCL-81) used in all assays were cultured in 75 cm^2^ flasks in Dulbecco’s Modified Eagle Medium (DMEM) (Gibco) supplemented with 2% fetal bovine serum (FBS), 100 IU/mL penicillin, and 100 µg/mL streptomycin. Cells were maintained at 37 °C, under 5% CO_2_ atmosphere. Mayaro virus (ATCC VR 66, lineage TR 4675) was initially isolated from Trinidad in 1954, and it was propagated in C6/36 cells (*A. albopictus*, ATCC CRL-1660) cultured in Leibowitz’s L15 medium, at 28 °C. Vero cells were seeded in 24-well plates (1 × 10^6^ cells) and incubated at 37 °C, 5% CO_2_ upon reaching approximately 80% of confluence for viral titration. Next, media was removed, and 100 µL of MAYV 10-fold dilutions were added onto cells. Plates were incubated at room temperature under agitation for 1 h to allow viral adsorption. Media was replaced by a new overlay solution (3% carboxymethylcellulose solution + DMEM, 1:2 ratio) and plates returned to incubation at 37 °C and 5% CO_2_ atmosphere for 48 h. Cytopathic effects were monitored. Plates were fixed with 10% formaldehyde solution and stained with 5% violet crystal solution. Lysis plaques were counted, and the mean was used to calculate plaque-forming units (PFU). Viral titers were expressed in PFU per milliliter (PFU/mL). 

### 2.3. Antiviral Screening Assay

Twenty-four cyclic ketones had their 50% cell cytotoxicity concentration determined (CC_50_) by our research group previously [37]. Briefly, cell cytotoxicity assay was performed using the MTT [(3-(4,5-dimethylthiazol-2-yl)-2,5-diphenyltetrazolium Bromide) Invitrogen™] assay for cell viability. Vero cells (5 × 10^3^ cells/well) were seeded on 96-well plates and incubated at 37 °C and 5% CO_2_ until reaching 70–80% of confluence. Different dilutions (1, 4, 8, 16, 32, 64, 128, 256 and 1000 µmol·L^−1^) of each compound were added to the cell monolayers followed by a 24 h incubation period. After the media was removed, an MTT solution diluted in fresh DMEM to a final concentration of 0.5 mg/mL was added to the cells, plates were returned to incubation at 37 °C for 4 h. After this period, media was replaced by 100 µL of DMSO to dissolve formazan crystals, and plates were placed on a platform shaker for homogenization. Absorbance was measured at 540 nm, and readings obtained were fitted into a non-linear regression to calculate CC_50_ values. The chemical structures of the ketones are depicted in Table 1.

Next, the compounds were screened for antiviral potential against MAYV using MTT assay as well. Vero cells (1 × 10^4^ cells/well) were seeded in 96-well plates until reaching 70–80% of confluence. the MAYV (MOI 1) was previously incubated with each compound for 1 h, then added to the cell monolayer and incubated for 1 h at 37 °C, 5% CO_2_ atmosphere. Subsequently, media was removed and 200 µL of fresh DMEM were added to the cells. Plates were incubated for 48 h, at 37 °C and 5% CO_2_. Plates were processed following the same MTT assay described above. Absorbance was measured at 540 nm, and values obtained were fitted into a non-linear regression. Negative control (cells with DMEM only) was considered 100% cell viability, and positive control (cells infected with MAYV) was considered 0% cell viability. Compounds that maintained cell viability equal to 50% or higher were selected for further experiments.

### 2.4. Viral Inhibition Assay

A viral inhibition assay was performed to determine the direct effect of a compound on viral particles. A MAYV dilution (MOI 1) was mixed in equal volumes to eight two-fold dilutions of each compound and incubated at 37 °C for 1 h. Next, dilutions were added to an 80% confluent Vero cell monolayer (1 × 10^6^ cells) in 24-well plates, then incubated on a platform shaker for 1 h. As positive control, a non-treated MAYV dilution (100 PFU/mL) was added to cells. After this period, media was removed and replaced by 1 mL of overlay solution (CMC 3% + DMEM), and plates were maintained at 37 °C and 5% CO_2_ atmosphere for 48 h. Plates were fixed in 10% formaldehyde solution and stained with 5% crystal violet solution. Lysis plaques were counted and fitted into a non-linear regression to calculate the dose-response viral activity reduction and 50% effective concentration (EC_50_) values.

### 2.5. Mechanisms of Action Assays

A series of lysis plaque assays were performed at different conditions to investigate whether the antiviral activity of xanthenodiones results from the interaction with the viral particle or with the cells. The following tests were performed according to Silva et al.’s previously described method [37] with some modifications. All assays were performed in 24-well plates using Vero cells at approximately 80% confluence.

#### 2.5.1. Pre-Treatment Assay

Compounds 1–24 (Table 1) were serially diluted two-fold, and each dilution was added to the cell monolayers. The plates were incubated at 37 °C for 3 h. Cells were washed with phosphate-buffered saline (PBS), a MAYV dilution (MOI 1) was added, and plates were returned to incubation at 37 °C for 1 h. Cells were rewashed with PBS and 1 mL of overlay solution was added. Plates were maintained at 37 °C for 48 h, followed by fixation and staining procedures as mentioned before. Lysis plaques were counted and compared to a positive control (non-treated MAYV-infected cells).

#### 2.5.2. Post-Treatment Assay

Cell monolayers were added a MAYV dilution (MOI 1) and incubated at 37 °C for 1 h under constant agitation for viral adsorption. Then, the viral suspension was removed, and two-fold serial dilutions of the evaluated compound were added to the cells. Plates returned to incubation at 37 °C for another 3 h. Media was aspirated and replaced with 1 mL of overlay solution. Cells were maintained at 37 °C for 48 h, followed by fixation and staining procedures.

#### 2.5.3. Viral Adsorption Inhibition Assay

A MAYV dilution (MOI 1) was added to equal volumes of two-fold serial dilutions of the evaluated compounds in microtubes. Cell monolayers were overlaid with 100 µL of the mixture and incubated at 4 °C for 1 h for viral adsorption under constant agitation. Subsequently, media was removed, and 1 mL of overlay solution was added. Cells were maintained at 37 °C and 5% CO_2_ for 48 h. Cells were fixed and stained.

### 2.6. Statistical Analysis

All data were obtained from three independent experiments with three replicates each. Results were analyzed using GraphPad Prism 8. Non-linear regression was used to calculate EC50 values. Multiple comparisons with One-way ANOVA determined the statistical significance of assays.

## 3. Results

### 3.1. Antiviral Screening

Twenty-four cyclic ketones (xanthenodiones 1–5, 8–23; tetraketones 6,7; diketone with an arylidene group 24) were screened for their antiviral activity by performing a cell viability assay using MTT. Vero cells infected with MAYV were treated with each compound and compared to a control group of non-treated MAYV-infected cells. The goal was to evaluate which compounds were able to maintain at least 50% of cell viability. Among compounds, eight of them (3, 5, 6, 7, 8, 9, 10, 12 and 15) were selected for further tests (as indicated in Figure 2).

### 3.2. Viral Inhibition Assay

In the next step, compounds were evaluated for their direct effect on viral particles. The 50% effective concentration was calculated via lysis-plaque assay as listed in Table 2 and their correspondent selectivity index (SI) was obtained from CC_50_/EC_50_ ratio [37]. Compound 9, a xanthenodione, exhibited the highest SI (15.8) and was therefore selected for the mechanism of action assays. At concentrations of 169.4 and 338.8 µmol·L^−1^, the substance was able to inhibit 60 and 80% of viral activity, respectively, compared to the positive control, as shown in Figure 3.

### 3.3. Mechanism of Action Assays

We conducted a series of plaque-forming tests to evaluate compound action in different stages of the viral cycle. First, we aimed to evaluate whether pre-treatment of cells with compound 9 could affect the development of the MAYV cycle. Therefore, Vero cells were pre-treated with different serial dilutions of compound 9 and incubated for 3 h, followed by the addition of MAYV suspension. As indicated in Figure 4A, compound 9 inhibited 77% viral activity at the highest concentration (338.8 µmol·L^−1^). Lower concentrations also impacted viral activity, although not the same extension. At 139.4 µmol·L^−1^, viral activity was inhibited by 66%, at concentrations 84.7 and 42.4 µmol·L^−1^ inhibition was approximately 40%.

Subsequently, we evaluated lysis plaque formation after MAYV-infected cells were treated with compound 9. The goal was to verify if the substance would interfere in intracellular stages of the viral cycle, such as replication, particle assembly, or budding. At the highest concentration (338.8 µmol·L^−1^), viral activity was inhibited by 71% (Figure 4B).

Compound 9 was also evaluated for adsorption inhibition properties. This assay added a MAYV dilution mixed with different serial dilutions of compound 9 to Vero cells kept at 4 °C. At this temperature, the virus can attach to cells but not internalize. Therefore, it would be possible to evaluate whether the substance acts directly on viral particles, interfering with the adsorption stage. As shown in Figure 4C, compound 9 did not inhibit the viral adsorption process at any concentration.

## 4. Discussion

The cyclic ketones are a group of compounds displaying several biological activities, including antiviral activities. Here, we investigated the anti-MAYV activity of twenty-four cyclic ketones (Table 1), initially by performing an antiviral screening assay to select compounds capable of maintaining cell viability at 50% or above. As a result, eight compounds (3, 5, 6, 7, 8, 9, 10, 12, and 15) were selected for further tests (Figure 2). Compound 9 was selected for a more detailed evaluation of its antiviral activity since it presented the highest SI (15.8) (Table 2).

In the pre-treatment assay, compound 9 inhibited viral activity in 70% of cells compared to non-treated MAYV-infected cells as the control (Figure 4A). This result possibly indicates that the compound acts on the cell surface or even intracellularly, impairing the viral cycle. A similar result was observed by Amorim et al. [22] when investigating the anti-MAYV activity of synthetic thienopyridine derivatives. The evaluated compound reduced viral activity by 50% when added to Vero cells 3 h before viral infection. However, the compound had a more substantial inhibitory effect when added with MAYV to the cell monolayers. 

The post-infection conducted with compound 9 resulted in a similar inhibitory effect of viral activity at the highest concentration (338.8 µmol·L^−1^) (Figure 4B) observed in the previous assay. It is possible to hypothesize that the substance can pass through cell membranes and target the viral cycle’s intracellular stages. Moreover, we tested if compound 9 could block viral attachment to cell receptors and consequent entrance in the host cell. However, the results obtained were not statistically significant (Figure 4C). Compound 9 had no inhibitory effects on the viral adsorption process. Similar results were observed by Ferreira et al. [23] in a study investigation of the anti-MAYV activity of epicatechin, a natural flavonoid isolated from *Salacia crassifolia*. Treatment of Vero cells in advance of MAYV inoculum resulted in a robust inhibitory effect. The anti-MAYV activity was also observed when treating cells and viral inoculum or alone with the substance. Like compound 9 in this study, epicatechin was unable to prevent viral adsorption. Comparing compound 9 and the flavonoid epicatechin, they present as common structural feature the presence of a pyran ring. In substance 9, the pyran ring is fused to a cyclohexen-2-one ring on both sides. The structure of epicatechin displays two phenolic rings linked through the heterocyclic pyran ring.

Our research group has previously described the antiviral activity of cyclic ketones against ZIKV, in which one compound–compound 24 in this study–showed a potent virucidal effect by interacting with viral envelope protein and blocking the viral adsorption process [37]. This fact points to the versatility of mechanisms in which cyclic ketones develop in different viral infection scenarios. However, it is necessary to continue investigating the mechanisms employed by these compounds against the Mayaro virus through more specific assays, accessing different stages of the viral cycle as well as evaluating antiviral activity of xanthenodione 9 using in vivo models. 

## 5. Conclusions

Mayaro virus is a neglected and emergent arbovirus that currently has no specific treatment or vaccines available. Therefore, the search for new therapeutic candidates is of utmost importance. In this study, a series of synthetic cyclic ketones were tested as antiviral candidates against the Mayaro virus. Compound 9 exhibited antiviral activity in pre- and post-treatment assays. Our group will further conduct further experiments in order to investigate specific aspects of the compound antiviral mechanisms.

## Figures and Tables

**Figure 1 viruses-13-02123-f001:**
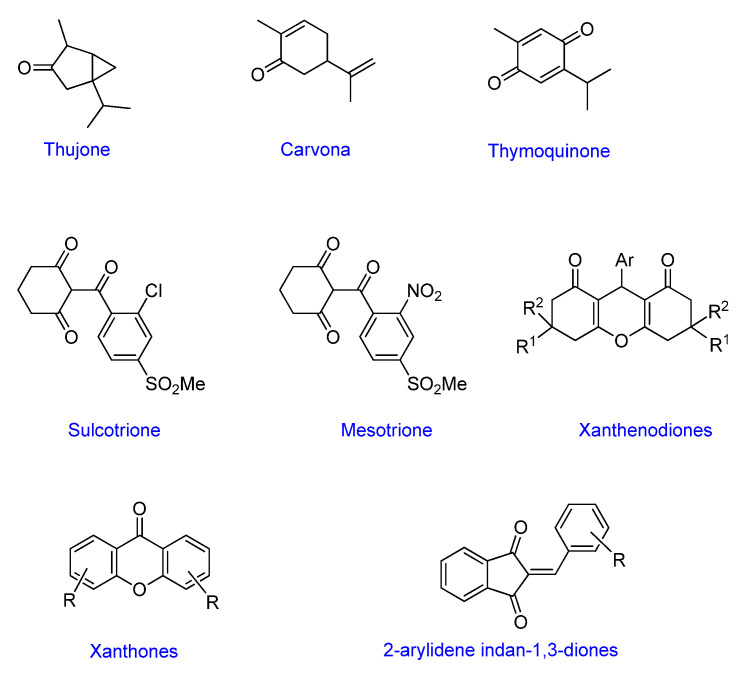
Structures of cyclic ketones.

**Figure 2 viruses-13-02123-f002:**
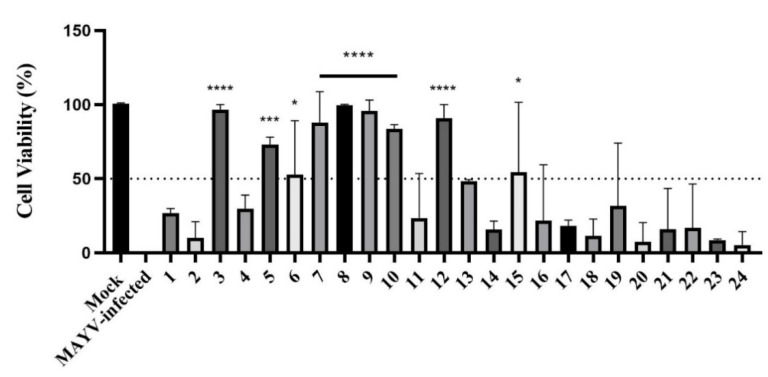
Antiviral screening of twenty-four cyclic ketones against MAYV. The MTT method was used to determine the cell viability of cells infected with MAYV and treated with the different compounds. For negative control, uninfected Vero cells (mock) were defined as 100% cell viability, and positive Control (MAYV-infected) was defined as 0%. **** *p* < 0.0001; *** *p* < 0.001; * *p* < 0.05.

**Figure 3 viruses-13-02123-f003:**
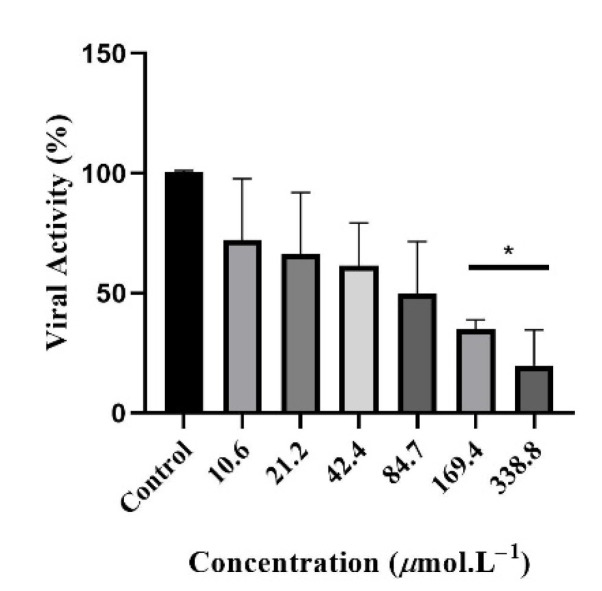
Viral inhibition assay of xanthenodione 9 against MAYV. Serial two-fold dilutions of compound 9 and a MAYV suspension (MOI 1) were mixed and incubated at 37 °C for 1 h. * *p* < 0.05.

**Figure 4 viruses-13-02123-f004:**
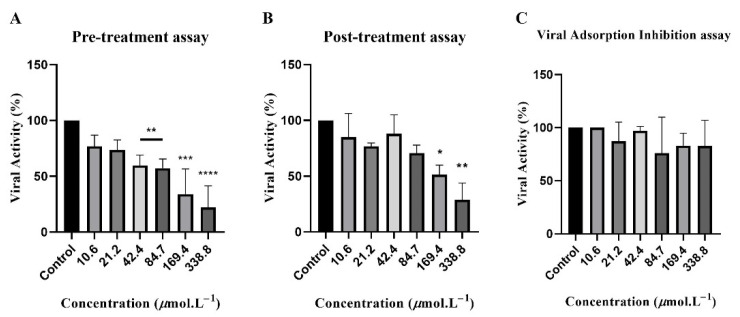
Mechanism of action assays. (**A**) Pre-treatment assay. Vero cells were pre-treated with compound 9 and subsequently infected with MAYV (MOI 1). Cells were incubated for 48 h, at 37 °C. (**B**) Post-treatment assay. MAYV-infected Vero cells were treated with different dilutions of compound 9. Cells were incubated for 48 h, at 37 °C. (**C**) Viral adsorption inhibition assay. A MAYV dilution (MOI 1) was mixed with different serial dilutions of the compound and added to Vero cells and incubated at 4 °C for 2 h and then moved to incubation for 48 h, at 37 °C. The viral activity was defined by lysis plaque counts and normalized to percentage considering control as 100% in all assays. The values obtained in Viral Adsorption Inhibition assay were non-significant. **** *p* < 0.0001; *** *p* < 0.001; ** *p* < 0.01; * *p* < 0.05.

**Table 1 viruses-13-02123-t001:** Chemical structures of cyclic ketones.

Compound	Structure	Compound	Structure
1	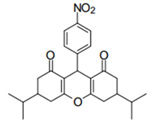	7	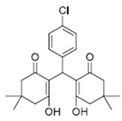
2	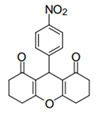	8	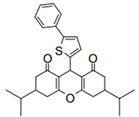
3	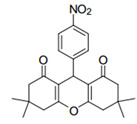	9	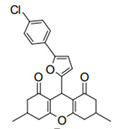
4	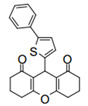	10	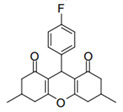
5	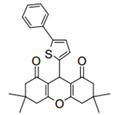	11	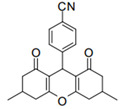
6	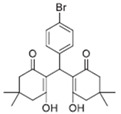	12	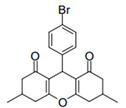
13	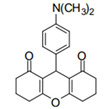	20	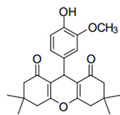
14	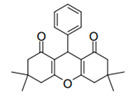	21	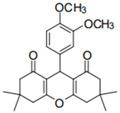
15	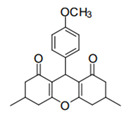	22	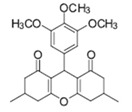
16	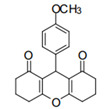	23	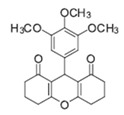
17	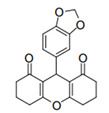	24	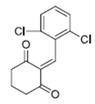
18	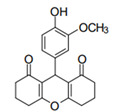		

**Table 2 viruses-13-02123-t002:** Cytotoxic concentration (CC_50_), effective concentration (EC_50_), and selectivity index (SI).

Compound	CC_50_ (µmol·L^−1^)	EC_50_ (µmol·L^−1^)	SI
3	292.6 ± 16.3	45.7 ± 13.4	6.4
5	27.6 ± 7.1	87.1 ± 32.3	0.3
7	91.2 ± 9.4	8.4 ± 2.0	10.8
8	336.8 ± 50.5	65.3 ± 15.8	5.2
9	338.8 ± 53.9	21.5 ± 6.5	15.8
10	324.1 ± 100.6	nd ^1^	nd
12	345.4 ± 47.4	77.1 ± 40.2	4.8
15	343.7 ± 86.3	nd	nd

^1^ nd: not determined.

## Data Availability

All relevant data regarding this study are included within this paper.

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
