# Peer review of "Evaluation of Antiviral Activity of Cyclic Ketones against Mayaro Virus"

_viruses, 2021, doi:10.3390/v13112123_

Round 1

Reviewer 1 Report

In the manuscript entitled: “Evaluation of antiviral of cyclic ketones against Mayaro virus”, Fernandes and collaborators, studied the anti-Mayaro effect of a series of synthetic ketones using a cytopathic effect-based assay as a primary screening. They found that the compound 9, a Xanthenodione showed the best selective index of activity and it was further analyzed. Although these results are interesting, the authors should clarify and improve several points.

  • In the section of Material and Methods, the authors should add more information about cell cytotoxicity assay, such as, time of incubation or doses tested with each compound.
  • The information provided in the Table 2 is unnecessary, since it appears in the Figure 2.
  • The results of mechanistic assays showed in Figure 4, 5 and 6 could be placed in a single figure with three panels. Thus, the subheadings 3.3.1 to 3.3.3 can be eliminated and the authors could describe their results in a single paragraph. Regarding to this, and the amount of data provided, this manuscript fit best as a brief report or communication than a full article.
  • In the legends of Figure 3, 4 and 6, the authors indicated that they used a MOI of 1 in each experiment. However, in Material and Methods section in lines 138, 142, 157, 163 and 170, the authors indicated that they used 100 or 200 PFU/mL. Please, clarify the exactly amount of virus used in each experiment.
  • Due the authors found an antiviral effect of compound 9 only at high doses, do you have information about the levels that this compound would reach in plasma in animal models? With regards to this, in the Discussion section describe the limitations of the study.

Author Response

Dear Reviewer 1,

          We would like to resubmit our Research Article entitled “Evaluation of antiviral activity of cyclic ketones against Mayaro virus (Manuscript ID: viruses-1429453), to be evaluated for publication in Viruses, special issue: "Antivirals for Arboviruses".

We are grateful for all comments and suggestions on this manuscript made by the reviewers. Modifications were performed in the manuscript taking into consideration every demand raised. We hope that this revised version is more suited to the journal.

Comments/suggestions made by the reviewers are responded below.

Reviewer 1:

In the manuscript entitled: “Evaluation of antiviral of cyclic ketones against Mayaro virus”, Fernandes and collaborators, studied the anti-Mayaro effect of a series of synthetic ketones using a cytopathic effect-based assay as a primary screening. They found that the compound 9, a Xanthenodione showed the best selective index of activity and it was further analyzed. Although these results are interesting, the authors should clarify and improve several points.

  1. In the section of Material and Methods, the authors should add more information about cell cytotoxicity assay, such as, time of incubation or doses tested with each compound.

          Cell cytotoxicity assay was previously described by Silva et al (2020) in another paper published by our research group. The following was added to clarify how cell cytotoxic assay was performed: “Twenty-four cyclic ketones had their 50% cell cytotoxicity concentration determined (CC50) by our research group previously [37]. Briefly, cell cytotoxicity assay was performed using the MTT [(3-(4,5-dimethylthiazol-2-yl)-2,5-diphenyltetrazolium Bromide) Invitrogen™] assay for cell viability. Vero cells (5x103 cells/well) were seeded on 96-well plates and incubated at 37 °C and 5% CO2 until reaching 70-80% of confluence. Different dilutions (1, 4, 8, 16, 32, 64, 128, 256 and 1000 µmol.L-1) of each compound were added to the cell monolayers followed by a 24 hour incubation period. After the media was removed, an MTT solution diluted in fresh DMEM to a final concentration of 0.5 mg/mL was added to the cells, plates were returned to incubation at 37 °C for 4 h. After this period, media was replaced by 100 µL of DMSO to dissolve formazan crystals, and plates were placed on a platform shaker for homogenization. Absorbance was measured at 540 nm, and readings obtained were fitted into a non-linear regression to calculate CC50 values. The chemical structures of the ketones are depicted in Table 1.”

          Next, description of antiviral screening assay was reformulated eliminating repetitive information contained in cell cytotoxicity assay: “Next, the compounds were screened for antiviral potential against MAYV using MTT assay as well. Vero cells (1x104 cells/well) were seeded in 96-well plates until reaching 70-80% of confluence. the MAYV (MOI 1) was previously incubated with each compound for 1 hour, then added to the cell monolayer and incubated for 1 hour at 37 °C, 5% CO2 atmosphere. Subsequently, media was removed and 200 µL of fresh DMEM were added to the cells. Plates were incubated for 48 h, at 37 °C and 5% CO2. Plates were processed following the same MTT assay described above. Absorbance was measured at 540 nm, and values obtained were fitted into a non-linear regression. Negative control (cells with DMEM only) was considered 100% cell viability, and positive control (cells infected with MAYV) was considered 0% cell viability. Compounds that maintained cell viability equal to 50% or higher were selected for further experiments.

  1. The information provided in the Table 2 is unnecessary, since it appears in the Figure 2.

          As suggested, Table 2 was removed from the manuscript since it contains the same information as Figure 2.

  1. The results of mechanistic assays showed in Figure 4, 5 and 6 could be placed in a single figure with three panels. Thus, the subheadings 3.3.1 to 3.3.3 can be eliminated and the authors could describe their results in a single paragraph. Regarding to this, and the amount of data provided, this manuscript fit best as a brief report or communication than a full article.

          As suggested, Figures 4, 5 and 6 were combined into a single figure with three panels. Also, modifications were done in the text regarding figure numbering: figures 4, 5 and 6 were assigned as Figure 4A, 4B and 4C.

          Content of subheadings 3.3.1 to 3.3.3 were combined into one single subheading 3.3 as well as respective results from this section (subheading 4.2).

  1. In the legends of Figure 3, 4 and 6, the authors indicated that they used a MOI of 1 in each experiment. However, in Material and Methods section in lines 138, 142, 157, 163 and 170, the authors indicated that they used 100 or 200 PFU/mL. Please, clarify the exactly amount of virus used in each experiment.

          Thank you for pointing out this mistake, concentration used in all assays was equivalent to MOI 1. Concentrations were corrected throughout the Material and Methods sections. In some assays the virus was added along with the compound and others the virus was added separately, that is the reason why viral dilution was 200 or 100 PFU, but the final concentration aimed for all assays was MOI 1.

  1. Due the authors found an antiviral effect of compound 9 only at high doses, do you have information about the levels that this compound would reach in plasma in animal models? With regards to this, in the Discussion section describe the limitations of the study.

          Unfortunately, we do not have information on that matter, our previous work (Silva, 2020) involving xanthenodiones did not cover this subject, in vivo studies were performed to evaluate viral load in brain tissue of one day-old mice after treatment with the xanthenodione 24. This is the first work demonstrating antiviral activity of compound 9. Certainly, more detailed studies both in vitro and in vivo regarding antiviral activity of compound 9 will be conducted in the future by our research group.

We hope the manuscript is now suitable for publication. If you need any additional information, please do not hesitate to contact me.

Best regards,

Prof. Dr. Sérgio Oliveira De Paula

Laboratory of Molecular Immunovirology

Department of General Biology

Federal University of Viçosa

CV: http://lattes.cnpq.br/8835491732653277

Reviewer 2 Report

I enjoyed reading this manuscript as it addresses antivirals against Mayaro virus.  MAYV is neglected, underdiagnosed, and under treated even though its just as serious as chikungunya and other endemic viruses. 

The manuscript was well written and study design and analysis sound.  I have a few minor comments.

Line 38: an "and" needs to be added before Una virus.

Line 69: an "at" should be placed between aiming and symptom.

Section 2.1: please add a brief summary of your ketone synthesis.

Section 2.2 FYI: a 1% concentration of crystal violet is sufficient to stain vero cells.

Line 125: place a coma at removed and delete the "and"

Line 152: please describe the modifications you used

Section 3.3: please add a justification why you chose to explore only compound 9. 

Table 2: please add standard error to the cell viability column 

Table 3: please add standard error to the CC50 and EC50 columns

Lines 265-266: please describe how the flavonoid relates to your ketone.  It will help your non-chemistry readers understand.

Author Response

Dear Reviewer 2,

          We would like to resubmit our Research Article entitled “Evaluation of antiviral activity of cyclic ketones against Mayaro virus (Manuscript ID: viruses-1429453), to be evaluated for publication in Viruses, special issue: "Antivirals for Arboviruses".

We are grateful for all comments and suggestions on this manuscript made by the reviewers. Modifications were performed in the manuscript taking into consideration every demand raised. We hope that this revised version is more suited to the journal.

Comments/suggestions made by the reviewers are responded below.

Reviewer 2:

I enjoyed reading this manuscript as it addresses antivirals against Mayaro virus.  MAYV is neglected, underdiagnosed, and under treated even though its just as serious as chikungunya and other endemic viruses. 

The manuscript was well written and study design and analysis sound.  I have a few minor comments.

  1. Line 38: an "and" needs to be added before Una virus.

     The sentence was corrected, thank you.

  1. Line 69: an "at" should be placed between aiming and symptom.

          The sentence was corrected, thank you.

  1. Section 2.1: please add a brief summary of your ketone synthesis.

          As requested, a brief summary of the ketones used in this work was added in line 97 as it follows: “A round-bottomed flask (25 mL) was charged with a 1,3-diketone (2.00 mmol), aldehyde (1.00 mmol) and ZrOCl2×8H2O (12.0 mg, 2 mol%). The mixture was stirred at 85 °C and the progress of the reaction was monitored by TLC analysis. After completion of the reaction, the mixture was cooled down to room temperature. Thereafter, it was added 50 mL of dichloromethane and the mixture was kept under stirring for about 30 minutes. Then, the catalyst, which is insoluble in dichloromethane, was separated by filtration. After that, 50 mL of ethanol was added to the filtrated and the system was kept undisturbed for crystallization. Detailed description of cyclic ketones synthesis was described previously [27, 37]”. The section title was also modified to “General procedure for preparation of cyclic ketones 1-24” as it suits more appropriately to the added content.

  1. Section 2.2 FYI: a 1% concentration of crystal violet is sufficient to stain vero cells.

          We appreciate the suggestion, future experiments will be carried out using this concentration.

  1. Line 125: place a coma at removed and delete the "and"

          Sentence was corrected, thank you, we appreciate all corrections regarding English grammar.

  1. Line 152: please describe the modifications you used

          Modifications in this protocol included different incubation periods. In Post-treatment assay (subheading 2.5.2), cells were incubated with the compound for 3 hours whereas in Silva et al’s work, cells and compounds were incubated for 2 hours. The second modification was regarding Viral Adsorption Inhibition assay (subheading 2.5.3). Silva et al’s protocol indicated a 30-minute incubation period of cells at 4 °C prior to addition of compounds and after that, maintaining cells at 4 °C for 2 hours. In this study, cells were incubated at 4 °C for 1 hour and only after compound addition to the plates.

  1. Section 3.3: please add a justification why you chose to explore only compound 9. 

          Compound 9 was selected because it presented the highest selectivity index.

  1. Table 2: please add standard error to the cell viability column

          Since Reviewer 1 suggested that Figure 2 and Table 2 contain repetitive information, we decided to remove Table 2 from the manuscript and maintain Figure 2, regarding the cell viability assay and presents both cell viability percentages and standard error bars.

  1. Table 3: please add standard error to the CC50 and EC50 columns

          Standard error was included in Table 3.

  1. Lines 265-266: please describe how the flavonoid relates to your ketone.  It will help your non-chemistry readers understand.

          The flavonoid is chemically related to the ketone as it presents similar chemical groups in its structure such as the pyran rings. The following was added to the text in line 279: “Comparing compound 9 and the flavonoid epicatechin, they present as common structural feature the presence of a pyran ring. In substance 9, the pyran ring is fused to a cyclohexen-2-one ring on both sides. The structure of epicatechin displays two phenolic rings linked through the heterocyclic pyran ring.”.

Reference:

Poly da Silva, Í.E.; Lopes da Silva, M.; Dias, R.S.; Santos, E.G.; Brangioni de Paula, M.C.; Silva de Oliveira, A.; Costa da Silveira Oliveira, A.F.; Marques de Oliveira, F.; Canedo da Silva, C.; Teixeira, R.R.; et al. Xanthenedione (and intermediates involved in their synthesis) inhibit Zika virus migration to the central nervous system in murine neonatal models. Microbes Infect. 2020, 22, 489–499, doi:10.1016/j.micinf.2020.04.007.

We hope the manuscript is now suitable for publication. If you need any additional information, please do not hesitate to contact me.

Best regards,

Prof. Dr. Sérgio Oliveira De Paula

Laboratory of Molecular Immunovirology

Department of General Biology

Federal University of Viçosa

CV: http://lattes.cnpq.br/8835491732653277